# PARL: Enhancing Diversity of Ensemble Networks to Resist Adversarial Attacks via Pairwise Adversarially Robust Loss Function

## Abstract

The security of Deep Learning classifiers is a critical field of study because of the existence of *adversarial attacks*. Such attacks usually rely on the principle of *transferability*, where an adversarial example crafted on a *surrogate classifier* tends to mislead the *target classifier* trained on the same dataset even if both classifiers have quite different architecture. Ensemble methods against adversarial attacks demonstrate that an adversarial example is less likely to mislead multiple classifiers in an ensemble having *diverse decision boundaries*. However, recent ensemble methods have either been shown to be vulnerable to stronger adversaries or shown to lack an end-to-end evaluation. This paper attempts to develop a new ensemble methodology that constructs multiple diverse classifiers using a **P**airwise **A**dversarially **R**obust **L**oss (PARL) function during the training procedure. PARL utilizes gradients of each layer with respect to input in every classifier within the ensemble simultaneously. The proposed training procedure enables PARL to achieve higher robustness against *black-box* transfer attacks compared to previous ensemble methods without adversely affecting the accuracy of clean examples. We also evaluate the robustness in the presence of *white-box* attacks, where adversarial examples are crafted using parameters of the target classifier. We present extensive experiments using standard image classification datasets like CIFAR-10 and CIFAR-100 trained using standard ResNet20 classifier against state-of-the-art adversarial attacks to demonstrate the robustness of the proposed ensemble methodology.

## 1 Introduction

Deep learning (DL) algorithms have seen rapid growth in recent years because of their unprecedented successes with near-human accuracies in a wide variety of challenging tasks starting from image classification (Szegedy et al., 2016), speech recognition (Amodei et al., 2016), natural language processing (Wu et al., 2016) to self-driving cars (Bojarski et al., 2016). While DL algorithms are extremely efficient in solving complicated decision-making tasks, they are vulnerable to well-crafted *adversarial examples* (slightly perturbed valid input with visually imperceptible noise) (Szegedy et al., 2014). The widely-studied phenomenon of adversarial examples among the research community has produced numerous attack methodologies with varied complexity and effective deceiving strategy (Goodfellow et al., 2015; Kurakin et al., 2017; Madry et al., 2018; Moosavi-Dezfooli et al., 2016; Papernot et al., 2017). An extensive range of defenses against such attacks has been proposed in the literature, which generally falls into two categories. The first category enhances the training strategy of deep learning models to make them less vulnerable to adversarial examples by training the models with different degrees of adversarially perturbed training data (Bastani et al., 2016; Huang et al., 2015; Jin et al., 2015; Zheng et al., 2016) or changing the training procedure like gradient masking, defensive distillation, etc. (Gu & Rigazio, 2015; Papernot et al., 2016; Rozsa et al., 2016; Shaham et al., 2015). However, developing such defenses has been shown to be extremely challenging. Carlini & Wagner (2017b) demonstrated that these defenses are not generalized for all varieties of adversarial attacks but are constrained to specific categories. The process of training with adversarially perturbed data is hard, often requires models with large capacity, and suffers from significant loss on clean example accuracy. Moreover, Athalye et al. (2018) demonstrated that the changes in training procedures provide a false sense of security. The second category intends to

detect adversarial examples by simply flagging them (Bhagoji et al., 2017; Feinman et al., 2017; Gong et al., 2017; Grosse et al., 2017; Metzen et al., 2017; Hendrycks & Gimpel, 2017; Li & Li, 2017). However, even detection of adversarial examples can be quite a complicated task. Carlini & Wagner (2017a) illustrated with several experimentations that these detection techniques could be efficiently bypassed by a strong adversary having partial or complete knowledge of the internal working procedure.

While all the approaches mentioned above deal with standalone classifiers, in this paper, we utilize the advantage of an ensemble of classifiers instead of a single standalone classifier to resist adversarial attacks. The notion of using diverse ensembles to increase the robustness of a classifier against adversarial examples has recently been explored in the research community. The primary motivation of using an ensemble-based defense is that if multiple classifiers with similar *decision boundaries* perform the same task, the *transferability* property of DL classifiers makes it easier for an adversary to mislead all the classifiers simultaneously using adversarial examples crafted on any of the classifiers. However, it will be difficult for an adversary to mislead multiple classifiers simultaneously if they have diverse decision boundaries. Strauss et al. (2017) used various ad-hoc techniques such as different random initializations, different neural network structures, bagging the input data, adding Gaussian noise while training for creating multiple diverse classifiers to form an ensemble. The resulting ensemble increases the robustness of the classification task even in the presence of adversarial examples. Tramèr et al. (2018) proposed *Ensemble Adversarial Training* that incorporates perturbed inputs transferred from other pre-trained models during adversarial training to decouple adversarial example generation from the parameters of the primary model. Grefenstette et al. (2018) demonstrated that ensembling two models and then adversarially training them incorporates more robustness in the classification task than single-model adversarial training and ensemble of two separately adversarially trained models. Kariyappa & Qureshi (2019) proposed *Diversity Training* of an ensemble of models with uncorrelated loss functions using *Gradient Alignment Loss* metric to reduce the dimension of adversarial sub-space shared between different models and increase the robustness of the classification task. Pang et al. (2019) proposed *Adaptive Diversity Promoting* regularizer to train an ensemble of classifiers that encourages the non-maximal predictions in each member in the ensemble to be mutually orthogonal, which degenerates the transferability that aids in resisting adversarial examples. Yang et al. (2020) proposed a methodology that isolates the adversarial vulnerability in each sub-model of an ensemble by distilling non-robust features. While all these works attempt to enhance the robustness of a classification task even in the presence of adversarial examples, Adam et al. (2018) proposed a stochastic method to add Variational Autoencoders between layers as a noise removal operator for creating combinatorial ensembles to limit the transferability and detect adversarial examples.

In order to enhance the robustness of a classification task and/or to detect adversarial examples, the ensemble-based approaches mentioned above employ different strategies while training the models. These ensembles either lack end-to-end evaluations for complicated datasets or lack evaluation against more aggressive attack scenarios like the methods discussed by Dong et al. (2018) and Liu et al. (2017), which demonstrate that adversarial examples misleading multiple models in an ensemble tend to be more transferable. *In this work, our primary objective is to propose a systematic approach to enhance the classification robustness of an ensemble of classifiers against adversarial examples by developing diversity in the decision boundaries among all the classifiers within that ensemble. The diversity is obtained by simultaneously considering the mutual dissimilarity in gradients of each layer with respect to input in every classifier while training them.* The diversity among the classifiers trained in such a way helps to degenerate the transferability of adversarial examples within the ensemble.

**Motivation behind the Proposed Approach and Contribution:** The intuition behind developing the proposed ensemble methodology is discussed in Figure 1, which shows a case study using classifiers trained on CIFAR-10 dataset without loss of generality. Figure 1a shows an input image of a 'frog'. Figure 1b shows the gradient of loss with respect to input[1] in a classifier $\mathcal{M}_{prim}$, and denoted as $\nabla_{prim}$. Figure 1c shows the gradient for another classifier $\mathcal{M}_{sim}$ with a *similar* decision boundary as $\mathcal{M}_{prim}$, and denoted as $\nabla_{sim}$. The classifier $\mathcal{M}_{sim}$ is trained using the same parameter settings as $\mathcal{M}_{prim}$ but with different random initialization. Figure 1d shows the gradients for a classifier $\mathcal{M}_{div}$ with a *not so similar* decision boundary compared to $\mathcal{M}_{prim}$, and denoted as

---

[1]Fundamental operation behind the creation of almost all adversarial examples

Figure 1: (a) Input image; (b) $\nabla_{prim}$: Gradient of loss in the primary classifier; (c) $\nabla_{sim}$: Gradient of loss in another classifier with *similar* decision boundaries; (d) $\nabla_{div}$: Gradient of loss in a classifier with *not so similar* decision boundaries but comparable accuracy; (e) Symbolic directions of all the gradients in higher dimensions. The gradients are computed with respect to the image shown in (a).

$\nabla_{div}$. The classifiers $\mathcal{M}_{prim}$, $\mathcal{M}_{sim}$, and $\mathcal{M}_{div}$ have similar classification accuracies. *The method for obtaining such classifiers is discussed later in this paper.* Figure 1e shows relative symbolic directions among all the aforementioned gradients in higher dimension. The directions between a pair of gradients are computed using *cosine similarity*. We can observe that $\nabla_{prim}$ and $\nabla_{sim}$ lead to almost in the same directions, aiding adversarial examples crafted on $\mathcal{M}_{prim}$ to *transfer* into $\mathcal{M}_{sim}$. However, adversarial examples crafted in $\mathcal{M}_{prim}$ will be difficult to transfer into $\mathcal{M}_{div}$ as the directions of $\nabla_{prim}$ and $\nabla_{div}$ are significantly different.

The principal motivation behind the proposed methodology is to introduce a constraint for reducing the cosine similarity of gradients among each classifier in an ensemble while training them simultaneously. Such a learning strategy with mutual cooperation intends to ensure that gradients between each pair of classifiers in the ensemble are as dissimilar as possible. We make the following contributions using the proposed ensemble training method:

- We propose a methodology to increase diversity in the decision boundaries among all the classifiers within an ensemble to degrade the transferability of adversarial examples.

- We propose a Pairwise Adversarially Robust Loss (PARL) function by utilizing the gradients of each layer with respect to input of every classifier within the ensemble simultaneously for training them to produce such varying decision boundaries.

- The proposed method can significantly improve the overall robustness of the ensemble against black-box transfer attacks without substantially impacting the clean example accuracy.

- We evaluated the robustness of PARL with extensive experiments using two standard image classification benchmark datasets on ResNet20 architecture against state-of-the-art adversarial attacks.

## 2 THREAT MODEL

We consider the following two threat models in this paper while generating adversarial examples.

- *Zero Knowledge Adversary* ($\mathcal{A}_{\mathcal{Z}}$): The adversary $\mathcal{A}_{\mathcal{Z}}$ does not have access to the target ensemble $\mathcal{M}_{\mathcal{T}}$ but has access to a surrogate ensemble $\mathcal{M}_{\mathcal{S}}$ trained with the same dataset. We term $\mathcal{A}_{\mathcal{Z}}$ as a *black-box* adversary. The adversary $\mathcal{A}_{\mathcal{Z}}$ crafts adversarial examples on $\mathcal{M}_{\mathcal{S}}$ and transfers to $\mathcal{M}_{\mathcal{T}}$.

- *Perfect Knowledge Adversary* ($\mathcal{A}_{\mathcal{P}}$): The adversary $\mathcal{A}_{\mathcal{P}}$ is a stronger than $\mathcal{A}_{\mathcal{Z}}$ who has access to the target ensemble $\mathcal{M}_{\mathcal{T}}$. We term $\mathcal{A}_{\mathcal{P}}$ as a *white-box* adversary. The adversary $\mathcal{A}_{\mathcal{P}}$ can generate adversarial examples on $\mathcal{M}_{\mathcal{T}}$ knowing the parameters used by all the networks within $\mathcal{M}_{\mathcal{T}}$.

## 3 OVERVIEW OF THE PROPOSED METHODOLOGY

In this section, we provide a brief description of the proposed methodology used in this paper to enhance classification robustness against adversarial examples using an ensemble of classifiers $\mathcal{M}_{\mathcal{T}}$. The ensemble consists of $\mathcal{N}$ neural network classifiers and denoted as $\mathcal{M}_{\mathcal{T}} = \bigcup_{i=1}^{\mathcal{N}} \mathcal{M}_i$. All the $\mathcal{M}_i$'s are trained simultaneously using the *Pairwise Adversarially Robust Loss* (PARL) function, which we discuss in detail in Section 4. The final decision for an input image on $\mathcal{M}_{\mathcal{T}}$ is decided based on the *majority voting* among all the classifiers. Formally, let us assume a test set of $t$ inputs $\{x_1, x_2, \ldots, x_t\}$ with respective ground truth labels as $\{y_1, y_2, \ldots, y_t\}$. The final decision of the ensemble $\mathcal{M}_{\mathcal{T}}$ for an input $x_j$ is defined as

$$\mathcal{C}(\mathcal{M}_{\mathcal{T}}, x_j) = majority\{\mathcal{M}_1(x_j), \mathcal{M}_2(x_j), \cdots, \mathcal{M}_{\mathcal{N}}(x_j)\}$$

$\mathcal{C}(\mathcal{M}_\mathcal{T}, x_j) = y_j$ for most $x_j$'s in an appropriately trained $\mathcal{M}_\mathcal{T}$. The primary argument behind the proposed ensemble method is that all $\mathcal{M}_i$'s have dissimilar decision boundaries but not significantly different accuracies. Hence, a clean example classified as class $\mathcal{C}_x$ in $\mathcal{M}_i$ will also be classified as $\mathcal{C}_x$ in most other $\mathcal{M}_j$'s (where $j = 1 \ldots \mathcal{N}$, $j \neq i$) within the ensemble with a high probability. Consequently, because of the diversity in decision boundaries between $\mathcal{M}_i$ and $\mathcal{M}_j$ (for $i = 1 \ldots \mathcal{N}$, $j = 1 \ldots \mathcal{N}$, and $i \neq j$), the adversarial examples generated by a *zero knowledge adversary* ($\mathcal{A}_\mathcal{Z}$) for a surrogate ensemble $\mathcal{M}_\mathcal{S}$ will have a different impact on each classifiers within the ensemble $\mathcal{M}_\mathcal{T}$, i.e., the *transferability* of adversarial examples will be challenging within the ensemble. A *perfect knowledge adversary* ($\mathcal{A}_\mathcal{P}$) can generate adversarial examples for the ensemble $\mathcal{M}_\mathcal{T}$. However, in this scenario, the input image perturbation will be in different directions because of the diversity in decision boundaries among all $\mathcal{M}_i$'s within the ensemble. The collective disparity in perturbation directions makes it challenging to craft adversarial examples for the ensemble. We evaluated our proposed methodology considering both the adversaries and presented the results in Section 5.

# 4 BUILDING THE ENSEMBLE NETWORKS USING PARL

In this section, we provide a detailed discussion on training an ensemble of neural networks using the proposed Pairwise Adversarially Robust Loss (PARL) function for increasing diversity among the networks. First, we define the basic terminologies used in the construction, followed by a detailed methodology of the training procedure.

## 4.1 BASIC TERMINOLOGIES USED IN THE CONSTRUCTION

Let us consider an ensemble $\mathcal{M}_\mathcal{T} = \bigcup_{i=1}^{\mathcal{N}} \mathcal{M}_i$, where $\mathcal{M}_i$ is the $i^{th}$ network in the ensemble. We assume that each of these networks has the same architecture with $\mathcal{L}_\mathcal{H}$ number of hidden layers. Let $\mathcal{J}_{\mathcal{M}_i}(\mathbf{x}, \mathbf{y})$ be the loss functions evaluating the amount of loss incurred by the network $\mathcal{M}_i$ for a data point $\mathbf{x}$, where $\mathbf{y}$ is the ground-truth label for $\mathbf{x}$. Let $\mathcal{F}_{\mathcal{M}_i}^{\mathcal{L}_k}(\mathbf{x})$ be the output of $k^{th}$ hidden layer on the network $\mathcal{M}_i$ for the data point $\mathbf{x}$. Let us assume $\mathcal{F}_{\mathcal{M}_i}^{\mathcal{L}_k}(\mathbf{x})$ has $\mathcal{D}(\mathcal{L}_k)$ number of output features. Let us consider $\nabla_x \mathcal{F}_{\mathcal{M}_i}^{\mathcal{L}_k}(\mathbf{x})$ denote the sum of gradients over each output feature of $k^{th}$ hidden layer with respect to input on the network $\mathcal{M}_i$ for the data point $\mathbf{x}$. Hence,

$$\nabla_x \mathcal{F}_{\mathcal{M}_i}^{\mathcal{L}_k}(\mathbf{x}) = \sum_{f=1}^{\mathcal{D}(\mathcal{L}_k)} \nabla_x [\mathcal{F}_{\mathcal{M}_i}^{\mathcal{L}_k}(\mathbf{x})]_f$$

where $\nabla_x [\mathcal{F}_{\mathcal{M}_i}^{\mathcal{L}_k}(\cdot)]_f$ is the gradient of the $f^{th}$ output feature of $k^{th}$ hidden layer on network $\mathcal{M}_i$ with respect to input for the data point $\mathbf{x}$. Let $\mathcal{X}$ be the training dataset containing $|\mathcal{X}|$ examples.

## 4.2 PAIRWISE ADVERSARIALLY ROBUST LOSS FUNCTION

The principal idea behind the proposed approach is to train an ensemble of neural networks such that the gradients of loss with respect to input in all the networks will be in different directions. The gradients represent the directions in which the input needs to be perturbed such that the loss of the network increases, helping to transfer adversarial examples. In this paper, we introduce the Pairwise Adversarially Robust Loss (PARL) function, which we will use to train the ensemble. The objective of PARL is to train the ensemble so that the gradients of loss lead to different directions in different networks for the same input example. Hence, the fundamental strategy is to make the gradients as dissimilar as possible while training all the networks. Since the gradient computation depends on all intermediate parameters of a network, we force the intermediate layers of all the networks within the ensemble to be dissimilar for producing enhanced diversity at each layer.

The pairwise similarity of gradients of the output of $k^{th}$ hidden layer with respect to input between the classifiers $\mathcal{M}_i$ and $\mathcal{M}_j$ for a particular data point $\mathbf{x}$ can be represented as

$$\mathcal{G}_{\mathcal{L}_k}^{(i,j)}(\mathbf{x}) = \frac{< \nabla_x \mathcal{F}_{\mathcal{M}_i}^{\mathcal{L}_k}(\mathbf{x}), \nabla_x \mathcal{F}_{\mathcal{M}_j}^{\mathcal{L}_k}(\mathbf{x}) >}{\|\nabla_x \mathcal{F}_{\mathcal{M}_i}^{\mathcal{L}_k}(\mathbf{x})\| \cdot \|\nabla_x \mathcal{F}_{\mathcal{M}_j}^{\mathcal{L}_k}(\mathbf{x})\|}$$

where $< a, b >$ represents the dot product between two vectors $a$ and $b$. The overall pairwise similarity between the classifiers $\mathcal{M}_i$ and $\mathcal{M}_j$ for a particular data point $\mathbf{x}$ considering $\mathcal{L}_\mathcal{H}$ hidden layers can be written as

$$\mathcal{G}^{(i,j)}(\mathbf{x}) = \sum_{k=1}^{\mathcal{H}} \mathcal{G}_{\mathcal{L}_k}^{(i,j)}(\mathbf{x})$$

Next, we define a penalty term $\mathcal{R}(\mathcal{M}_i, \mathcal{M}_j)$ for all the training examples in $\mathcal{X}$ to pairwise train the models $\mathcal{M}_i$ and $\mathcal{M}_j$ as

$$\mathcal{R}(\mathcal{M}_i, \mathcal{M}_j) = \frac{1}{|\mathcal{X}|} \sum_{\mathbf{x} \in \mathcal{X}} \mathcal{G}^{(i,j)}(\mathbf{x})$$

We can observe that $\mathcal{R}$ computes average pairwise similarity for all the training examples. Now, for network $\mathcal{M}_i$ and $\mathcal{M}_j$, if all the gradients with respect to input for each training example are in the same direction, value of $\mathcal{R}$ will be close to 1, indicating similarity in decision boundaries. The value of $\mathcal{R}$ will gradually decrease as the relative angle between the pair of gradients increases in higher dimension. Hence, the objective of diversity training using PARL is to reduce the value of $\mathcal{R}$. Thus, we add $\mathcal{R}$ to the loss function as a penalty parameter to penalize the training for a large $\mathcal{R}$ value.

In the ensemble $\mathcal{M}_\mathcal{T}$, we compute the $\mathcal{R}$ values for each distinct pair of $\mathcal{M}_i$ and $\mathcal{M}_j$ in order to enforce diversity between each pair of classifiers. We define PARL to train the ensemble $\mathcal{M}_\mathcal{T}$ as

$$PARL(\mathcal{M}_\mathcal{T}) = \gamma_1 \cdot \frac{1}{|\mathcal{X}|} \sum_{\mathbf{x} \in \mathcal{X}} \sum_{i=1}^{\mathcal{N}} J_{\mathcal{M}_i}(\mathbf{x}, \mathbf{y}) + \gamma_2 \cdot \sum_{1 \leq i < j \leq \mathcal{N}} \mathcal{R}(\mathcal{M}_i, \mathcal{M}_j) \tag{1}$$

where $\gamma_1$ and $\gamma_2$ are hyperparameters controlling the accuracy-robustness trade-off. A higher value of $\gamma_1$ and a lower value of $\gamma_2$ helps to learn the models with good accuracy but is less robust against adversarial attacks. However, a lower value of $\gamma_1$ and a higher value of $\gamma_2$ makes the models more robust against adversarial attacks but with a compromise in overall accuracy.

One may note that the inclusion of the penalty values for each distinct pair of classifiers within the ensemble to compute PARL has one fundamental advantage. If we do not include the pair $(\mathcal{M}_a, \mathcal{M}_b)$ in the PARL computation, the training will continue without any diversity restrictions between $\mathcal{M}_a$ and $\mathcal{M}_b$. Consequently, $\mathcal{M}_a$ and $\mathcal{M}_b$ will produce similar decision boundaries, thereby increasing the likelihood of adversarial transferability between $\mathcal{M}_a$ and $\mathcal{M}_b$, affecting the robustness of the ensemble $\mathcal{M}_\mathcal{T}$. One may also note that the number of gradient computations in an efficient implementation of PARL is linearly proportional to the number of classifiers in the ensemble. The gradients for each classifier are computed once and are reused to compute $\mathcal{R}$ values for each pair of classifiers. The reuse of gradients protects the implementation from the exponential computational overhead due to pairwise similarity computation.

## 5 EXPERIMENTAL EVALUATION

### 5.1 EVALUATION CONFIGURATIONS

We consider an ensemble of three standard ResNet20 (He et al., 2016) architecture for all the ensembles used in this paper. We consider two standard image classification datasets for our evaluation, namely CIFAR-10 (Krizhevsky et al., 2009) and CIFAR-100 (Krizhevsky et al., 2009). We consider two scenarios for the evaluation:

- *Unprotected Ensemble*: A baseline ensemble ($ENS_\mathcal{U}$) of ResNet20 architectures without any countermeasure against adversarial attacks.
- *Protected Ensemble*: An ensemble ($ENS_\mathcal{Z}$) of ResNet20 architectures, where $\mathcal{Z}$ is the countermeasure used to design the ensemble. In our evaluation we have considered three previously proposed countermeasures to compare the performance of PARL. We denote the ensembles $ENS_{ADP}$, $ENS_{GAL}$, and $ENS_{DVERGE}$ to be the ensembles trained with the methods proposed by Pang et al. (2019), Kariyappa & Qureshi (2019), and Yang et al. (2020), respectively. The ensemble trained with our proposed method is denoted as $ENS_{PARL}$.

We use the *adam optimization* (Kingma & Ba, 2015) to train all the ensembles with adaptive learning rate starting from 0.001. We dynamically generate a augmented dataset using random shifts, flips and crops to train both CIFAR-10 and CIFAR-100. We use the default hyperparameter settings mentioned in the respective papers for $ENS_{ADP}$, $ENS_{GAL}$, and $ENS_{DVERGE}$[2]. We use $\gamma_1 = $

---

[2]We implemented $ENS_{GAL}$ following the approach mentioned in the paper. Whereas, we adopted the official GitHub repositories for $ENS_{ADP}$ and $ENS_{DVERGE}$ implementation.

1.0, $\gamma_2 = 0.5$, and *categorical crossentropy* loss for $\mathcal{J}_{\mathcal{M}_i}(\cdot)$ (ref. Equation (1)) for $ENS_{PARL}$. We enforce diversity among all the classifiers in $ENS_{PARL}$ for the first seven convolution layers[3].

We consider four state-of-the-art untargeted adversarial attacks Fast Gradient Sign Method (FGSM) (Goodfellow et al., 2015), Basic Iterative Method (BIM) (Kurakin et al., 2017), Momentum Iterative Method (MIM) (Dong et al., 2018), and Projected Gradient Descent (PGD) (Madry et al., 2018) for crafting adversarial examples. A brief overview on crafting adversarial examples using these attack methodologies are discussed in Appendix A. We consider 50 steps for generating adversarial examples using the iterative methods BIM, MIM, and PGD with the step size of $\epsilon/5$, where $\epsilon$ is the attack strength. We consider the moment decay factor as 0.01 for MIM. We use 10 different random restarts for PGD to generate multiple instances of adversarial examples for an impartial evaluation. We use $\epsilon = 0.01, 0.02, 0.03, 0.04, 0.05, 0.06$, and $0.07$ for generating adversarial examples of different strengths. We use `CleverHans v2.1.0` library (Papernot et al., 2018) to generate all the adversarial examples.

In order to evaluate against a stronger adversarial setting for $\mathcal{A}_{\mathcal{Z}}$, we train a black-box surrogate ensemble and generate adversarial examples from the surrogate ensemble instead of a standalone classifier. As also mentioned previously, adversarial examples that mislead multiple models in an ensemble tend to be more transferable (Dong et al., 2018; Liu et al., 2017). *All the results in the subsequent discussions are reported by taking average value over three independent runs.*

## 5.2 ANALYSING THE DIVERSITY

The primary objective of PARL is to increase the diversity among all the classifiers within an ensemble. In order to analyze the diversity of different classifiers trained using PARL, we use Linear Central Kernel Alignment (CKA) analysis proposed by Kornblith et al. (2019). The CKA metric, which lies between $[0, 1]$, measures the similarity between decision boundaries represented by a pair of neural networks. A higher CKA value between two neural networks indicates a significant similarity in decision boundary representations, which implies good transferability of adversarial examples. We present an analysis on layer-wise CKA values for each pair of classifiers within the ensemble $ENS_{\mathcal{U}}$ and $ENS_{PARL}$ trained with CIFAR-10 and CIFAR-100 in Figure 2 to show the effect of PARL on diversity. We can observe that each pair of models in $ENS_{\mathcal{U}}$ show a significant similarity at every layer. However, since $ENS_{PARL}$ is trained by restricting the first seven

---

[3]We observed that PARL performs better than previously proposed approaches against adversarial attacks with high accuracy on clean examples by enforcing diversity in the first seven convolution layers. We present an Ablation Study by varying the number of layers utilized for diversity training using PARL in Section 5.4

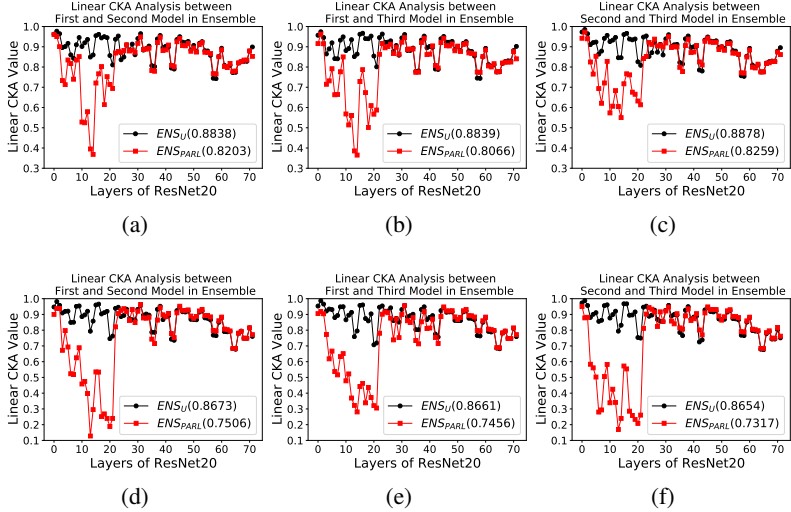

Figure 2: Layer-wise linear CKA values between each pair of classifiers in $ENS_{\mathcal{U}}$ and $ENS_{PARL}$ trained with CIFAR-10 [(a), (b), (c)] and CIFAR-100 [(d), (e), (f)] datasets showing the similarities at each layer. The value inside the braces within the corresponding figure legends represent the overall average Linear CKA values between each pair of classifiers.

Table 1: Ensemble classification accuracy (%) for CIFAR-10 and CIFAR-100 on clean examples as well as adversarially perturbed images with attack strength $\epsilon = 0.07$ for different adversarial attacks.

|  |  | Clean Example | FGSM | BIM | MIM | PGD |
|---|---|---|---|---|---|---|
| $ENS_{\mathcal{U}}$ | CIFAR-10 | 93.11 | 21.87 | 7.47 | 7.27 | 1.57 |
|  | CIFAR-100 | 70.88 | 7.13 | 7.17 | 3.93 | 5.39 |
| $ENS_{ADP}$ | CIFAR-10 | 92.99 | 23.53 | 8.13 | 7.53 | 3.08 |
|  | CIFAR-100 | 70.01 | 8.23 | 10.07 | 5.83 | 8.72 |
| $ENS_{GAL}$ | CIFAR-10 | 91.22 | 22.8 | 9.3 | 8.33 | 7.64 |
|  | CIFAR-100 | * | * | * | * | * |
| $ENS_{DVERGE}$ | CIFAR-10 | 91.73 | 28.88 | 11.78 | 10.55 | 9.68 |
|  | CIFAR-100 | * | * | * | * | * |
| $ENS_{PARL}$ | CIFAR-10 | 91.09 | 28.37 | 20.8 | 16.17 | 15.65 |
|  | CIFAR-100 | 67.52 | 12.73 | 18.97 | 10.01 | 21.49 |

\* Did not consider CIFAR-100 dataset for evaluation

convolution layers, we can observe a considerable decline in the CKA values at the initial layers. The observation is expected as $PARL$ imposes layer-wise diversity in its formulation, as discussed previously in Section 4. The overall average Linear CKA values between each pair of models in Figure 2 are mentioned inside braces within the corresponding figure legends, which signifies that the classifiers within an ensemble trained using PARL shows a higher overall dissimilarity than the unprotected baseline ensemble. In the subsequent discussions, we analyze the effect of the observed diversity on the performance of $ENS_{PARL}$ against adversarial examples.

### 5.3 ROBUSTNESS EVALUATION OF PARL

**Performance in the presence of $\mathcal{A}_{\mathcal{Z}}$:** We evaluate the robustness of all the ensembles discussed in Section 5.1 considering both CIFAR-10 and CIFAR-100 where the attackers cannot access the model parameters and rely on surrogate models to generate transferable adversarial examples. Under such a black-box scenario, we use one hold-out ensemble with three ResNet20 models as the surrogate model. We randomly select 1000 test samples and evaluate the performance of black-box transfer attacks for all the ensembles across a wide range of attack strength $\epsilon$. We present the result for the attack strength ($\epsilon = 0.07$) in Table 1 along with clean example accuracies. The methods presented by Kariyappa & Qureshi (2019) and Yang et al. (2020) neither evaluated their approach on a more complicated CIFAR-100 dataset nor discussed any optimal hyperparameter settings regarding their proposed algorithms for potential evaluation. In our robustness evaluations, we consider all the ensembles discussed in Section 5.1 for CIFAR-10. In contrast, we only consider $ENS_{\mathcal{U}}$ and $ENS_{ADP}$ to compare the performance of $ENS_{PARL}$ for CIFAR-100. We can observe from Table 1 that the training using PARL does not adversely affect the ensemble accuracy on clean examples compared to other previously proposed methodologies. However, it provides better robustness against black-box transfer attacks for almost every scenario.

A more detailed performance evaluation considering all the attack strengths are presented in Figure 3. For CIFAR-10 evaluation (ref. Figure 3a - Figure 3d), we can observe that $ENS_{PARL}$ performs at par with $ENS_{DVERGE}$ considering FGSM. However, for stronger iterative attacks like BIM, MIM, and PGD, $ENS_{PARL}$ outperforms other methodologies with higher accuracy for large attack strengths. For CIFAR-100 evaluation (ref. Figure 3e - Figure 3h), we can observe that $ENS_{PARL}$ performs better than both $ENS_{\mathcal{U}}$ and $ENS_{ADP}$ for all scenarios.

**Performance in the Presence of $\mathcal{A}_{\mathcal{P}}$:** Next, we evaluate the robustness of ensembles when the attacker has complete access to the model parameters. Under such a white-box scenario, we craft adversarial examples from the target ensemble itself. We consider the same attack methodologies and settings as discussed in Section 5.1. We randomly select 1000 test samples and evaluate white-box attacks for all the ensembles across a wide range of attack strength $\epsilon$. We present the results for CIFAR-10 and CIFAR-100 in Figure 4. For CIFAR-10 evaluation (ref. Figure 4a - Figure 4d), we can observe that $ENS_{PARL}$ performs marginally better than the other ensembles. For CIFAR-100

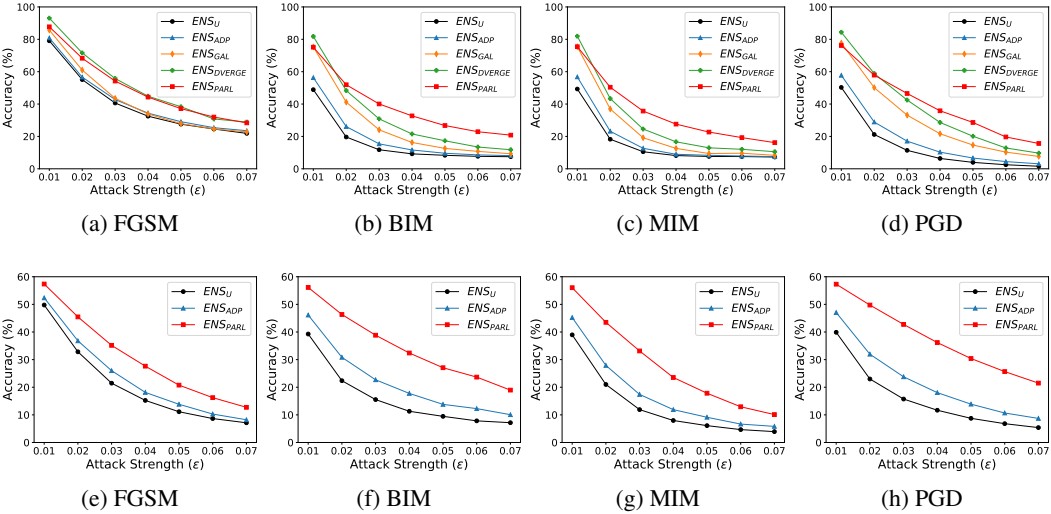

Figure 3: Ensemble classification accuracy (%) v.s. Attack Strength ($\epsilon$) against different black-box transfer attacks generated from surrogate ensemble for CIFAR-10 [(a), (b), (c), (d)] and CIFAR-100 [(e), (f), (g), (h)] datasets.

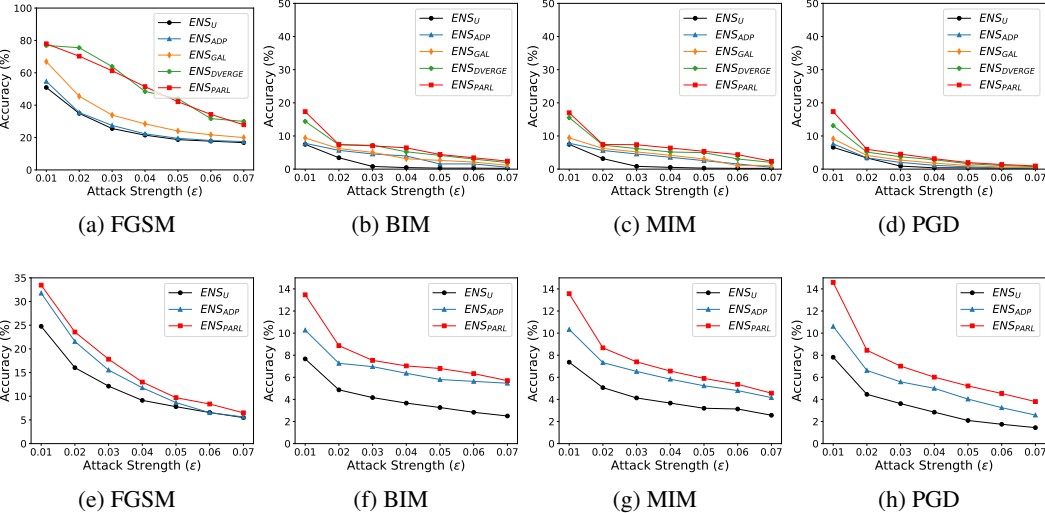

Figure 4: Ensemble classification accuracy (%) v.s. Attack Strength ($\epsilon$) against different white-box attacks for CIFAR-10 [(a), (b), (c), (d)] and CIFAR-100 [(e), (f), (g), (h)] datasets.

evaluation (ref. Figure 4e - Figure 4h), we can observe that $ENS_{PARL}$ performs better than both $ENS_{\mathcal{U}}$ and $ENS_{ADP}$ for all scenarios. Although PARL achieves the highest robustness among all the previous ensemble methods for black-box transfer attacks, its robustness against white-box attacks is still quite low. The result is expected as the objective of PARL is to increase the diversity of an ensemble against adversarial vulnerability rather than entirely eliminate it. In other words, adversarial vulnerability inevitably exists within the ensemble and can be captured by attacks with white-box access. One straightforward way to improve the robustness of ensembles against such vulnerability is to augment PARL with adversarial training (Madry et al., 2018), which we look forward to as a future research direction.

## 5.4 ABLATION STUDY

In all the previous evaluations, we consider $ENS_{PARL}$ by enforcing diversity in the first seven convolution layers for all the classifiers during ensemble training using PARL. In this section, we provide an ablation study by analyzing a varying number of convolution layers considered for the

Table 2: Ensemble classification accuracy (%) on the test set for CIFAR-10 and CIFAR-100. The numbers in the first row after slash denote the number of convolution layers used to enforce diversity

|  | $ENS_{PARL/7}$ | $ENS_{PARL/5}$ | $ENS_{PARL/3}$ |
|---|---|---|---|
| CIFAR-10 | 91.09 | 91.18 | 92.45 |
| CIFAR-100 | 67.52 | 67.54 | 68.81 |

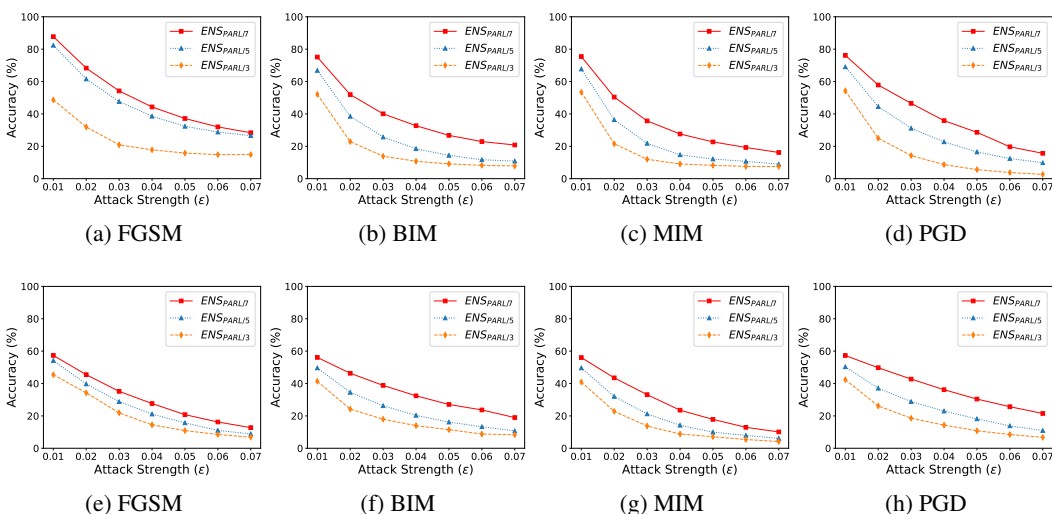

|  |  |  |  |
|---|---|---|---|
| (a) FGSM | (b) BIM | (c) MIM | (d) PGD |
| (e) FGSM | (f) BIM | (g) MIM | (h) PGD |

Figure 5: Ensemble classification accuracy (%) v.s. Attack Strength ($\epsilon$) against different black-box transfer attacks generated from surrogate ensemble for CIFAR-10 [(a), (b), (c), (d)] and CIFAR-100 [(e), (f), (g), (h)] datasets. The numbers in the figure legends after slash denote the number of convolution layers used to enforce diversity

diversity training. We consider three ensembles $ENS_{PARL/7}$, $ENS_{PARL/5}$, and $ENS_{PARL/3}$ for this study, where $ENS_{PARL/N}$ denotes that the first $N$ convolution layers are used for enforcing the diversity. We consider the same evaluation configurations as discussed in Section 5.1. The accuracies of all the ensembles on clean examples considering both CIFAR-10 and CIFAR-100 are mentioned in Table 2. We can observe that as fewer restrictions are imposed, the overall ensemble accuracy increases, which is expected and can be followed from Equation (1). A detailed analysis on the diversity achieved by all these ensembles in terms of Linear CKA metric is provided in Appendix B for interested readers.

Next, we evaluate the robustness of these ensembles against black-box transfer attacks for the evaluation configurations discussed in Section 5.1 and provide the results in Figure 5. We can observe that though the ensemble accuracy on clean examples increases with fewer layer restrictions, the robustness against black-box transfer attacks decreases significantly. The results present an interesting trade-off between accuracy and robustness in terms of number of layers considered while computing PARL (ref. Equation (1)).

## 6 CONCLUSION

In this paper, we propose an approach to enhance the classification robustness of an ensemble against adversarial attacks by developing diversity in the decision boundaries among all the classifiers within the ensemble. The ensemble network is constructed by the proposed Pairwise Adversarially Robust Loss function utilizing the gradients of each layer with respect to input in all the networks simultaneously. The experimental results show that the proposed method can significantly improve the overall robustness of the ensemble against state-of-the-art black-box transfer attacks without substantially impacting the clean example accuracy. Combining the technique with adversarial training and exploring different efficient methods to construct networks with diverse decision boundaries adhering to the principle outlined in the paper can be interesting future research directions.

## REPRODUCIBILITY STATEMENT

We have uploaded the codes as a supplementary material along with a README file for reproducing the results reported in this paper.

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

## A  BRIEF OVERVIEW OF ADVERSARIAL EXAMPLE GENERATION

Let us consider a benign data point $x \in \mathbb{R}^d$, classified into class $\mathcal{C}_i$ by a classifier $\mathcal{M}$. An untargeted adversarial attack tries to add visually imperceptible perturbation $\eta \in \mathbb{R}^d$ to $x$ and creates a new data point $x_{adv} = x + \eta$ such that $\mathcal{M}$ misclassifies $x_{adv}$ into another class $\mathcal{C}_j$ other than $\mathcal{C}_i$. The imperceptibility is enforced by restricting the $l_\infty$-norm of the perturbation $\eta$ to be below a threshold $\epsilon$, i.e., $\|\eta\|_\infty \leq \epsilon$ (Goodfellow et al., 2015; Madry et al., 2018). We term $\epsilon$ as the *attack strength*.

An adversary can craft adversarial examples by considering the loss function of a model, assuming that the adversary has full access to the model parameters. Let $\mathcal{J}(\theta, x, y)$ denote the loss function of the model, where $\theta$, $x$, and $y$ represent the model parameters, benign input, and corresponding label, respectively. The goal of the adversary is to generate adversarial example $x_{adv}$ such that the loss of the model is maximized while ensuring that the magnitude of the perturbation is upper bounded by $\eta$. Hence, $\mathcal{J}(\theta, x_{adv}, y) > \mathcal{J}(\theta, x, y)$ adhering to the constraint $\|x_{adv} - x\|_\infty \leq \epsilon$. Several methods have been proposed in the literature to solve this constrained optimization problem. We discuss the methods used in the evaluation of our proposed approach.

**Fast Gradient Sign Method (FGSM):** This attack is proposed by Goodfellow et al. (2015). The adversarial example is crafted using the following equation

$$x_{adv} = x + \epsilon \cdot sign(\nabla_x J(\theta, x, y))$$

where $\nabla_x J(\theta, x, y)$ denote the gradient of loss with respect to input.

**Basic Iterative Method (BIM):** This attack is proposed by Kurakin et al. (2017). The adversarial example is crafted iteratively using the following equation

$$x^{(k+1)} = x^{(k)} + clip_\epsilon(\alpha \cdot sign(\nabla_x J(\theta, x^{(k)}, y)))$$

where $x^{(0)}$ is the benign example. If $r$ is the total number of attack iteration, $x_{adv} = x^{(r)}$. The parameter $\alpha$ is a small step size usually chosen as $\epsilon/r$. The function $clip_{\epsilon}(\cdot)$ is used to generate adversarial examples within $\epsilon$-ball of the original image $x^{(0)}$.

**Moment Iterative Method (MIM):** This attack is proposed by Dong et al. (2018), which won the NeurIPS 2017 Adversarial Competition. This attack is a variant of BIM that crafts adversarial examples iteratively using the following equations

$$g^{(k)} = \nabla_x J(\theta, x^{(k)}, y)$$
$$g^{(k+1)} = \mu \cdot g^{(k)} + \frac{g^{(k)}}{\|g^{(k)}\|_1}$$
$$x^{(k+1)} = x^{(k)} + clip_{\epsilon}(\alpha \cdot sign(g^{(k)}))$$

where $\mu$ is termed as the decay factor.

**Projected Gradient Sign Method (PGD):** This attack is proposed by Madry et al. (2018). This attack is a variant of BIM with same adversarial example generation process except for $x^{(0)}$ is a randomly perturbed image in the $\epsilon$ neighborhood of the original image.

## B    DIVERSITY RESULTS FOR ABLATION STUDY

In this section, we present an analysis on layer-wise CKA values for each pair of classifiers within the ensemble $ENS_{PARL/7}$, $ENS_{PARL/5}$, and $ENS_{PARL/3}$ trained with both CIFAR-10 and CIFAR-100 datasets. The layer-wise CKA values are shown in Figure 6 to demonstrate the effect of PARL on diversity. We can observe a more significant decline in the CKA values at the initial layers for $ENS_{PARL/7}$ as compared to $ENS_{PARL/5}$ and $ENS_{PARL/3}$, which is expected as $ENS_{PARL/7}$ is trained by restricting more convolution layers. We can also observe that each pair of classifiers show more overall diversity in $ENS_{PARL/7}$ than in $ENS_{PARL/5}$ and $ENS_{PARL/3}$. The overall CKA values are mentioned inside braces within the corresponding figure legends.

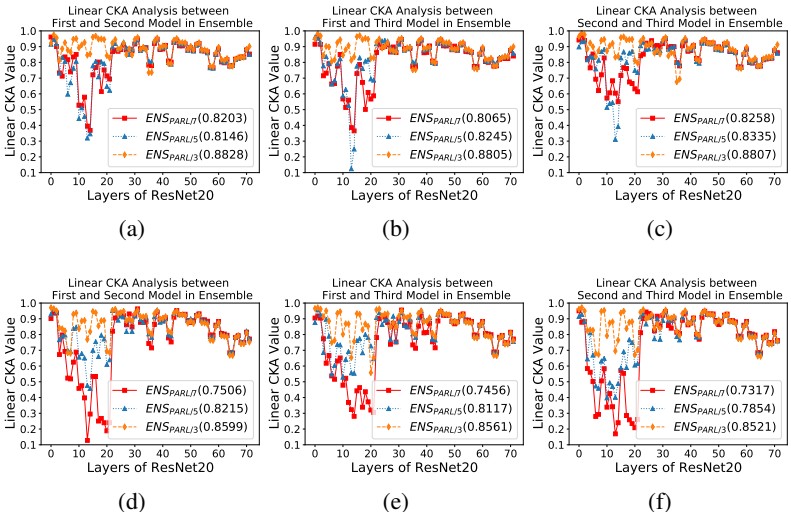

Figure 6: Layer-wise linear CKA values between each pair of classifiers in $ENS_{PARL/7}$, $ENS_{PARL/5}$, and $ENS_{PARL/3}$ trained with CIFAR-10 [(a), (b), (c)] and CIFAR-100 [(d), (e), (f)] datasets showing the similarities at each layer. The numbers in the figure legends after slash denote the number of convolution layers used to enforce diversity. The value inside the braces within the corresponding figure legends represent the overall average Linear CKA values between each pair of classifiers.

