# OpenReview forum: "PARL: Enhancing Diversity of Ensemble Networks to Resist Adversarial Attacks via Pairwise Adversarially Robust Loss Function"
_ICLR.cc/2022/Conference — ICLR 2022 Submitted_

### Official Review · Reviewer_BGda · 2021-10-28

**Correctness:** 3
**Technical Novelty And Significance:** 3
**Empirical Novelty And Significance:** 2
**Recommendation:** 3
**Confidence:** 4

**Main Review:**

The main idea of training diverse components makes sense.
Some details need better explanation or justification:
* First paragraph of section 4.1: it seems coarse to sum the gradients across a layer, and lots of information can get lost.
* It seems the equation at top of page 5 should be divided by L_H, because the text says that R(M_i,M_j) is at most 1.
* Why only compute the loss term on the first seven layers? In particular, Figure 5 suggests that the more the better. And why choose early layers?

The main weakness is that the results are not strong.
* According to Figure 3, the defense against transfer attack on CIFAR-10 is weaker than (Madry et al. 2018)'s white-box defense. Although the proposed method has a higher standard accuracy on clean inputs than (Madry et al. 2018), it's not good to have a weaker robustness under transfer attacks than a competitor's white-box robustness.
* The comparison is against three ensemble methods. There are other non-ensemble methods against transfer attacks that should be included in comparison, for example
https://arxiv.org/pdf/1705.09064


**Summary Of The Paper:**

This paper addresses the problem of L_inf transfer attacks.
The proposed method is an ensemble that is trained with an extra loss term that measures the pairwise and layerwise similarity of components. This loss term is the average cosine similarity between gradients with respect to the input.
At inference time, the overall prediction is by voting.
Experiments show better defense against L_inf transfer attacks than three other ensemble methods.

**Summary Of The Review:**

The idea makes sense, but results are weak.

---

### Official Review · Reviewer_SqzR · 2021-10-30

**Correctness:** 4
**Technical Novelty And Significance:** 3
**Empirical Novelty And Significance:** 3
**Recommendation:** 6
**Confidence:** 5

**Main Review:**

I mainly have the following comments/questions for the authors.

### Comparing with GAL
1. The high-level idea of the proposed PARL is "to train an ensemble of neural networks such that the gradients of loss with respect to input in all the networks will be in different directions". Given that this is the same as the idea of GAL, one of the considered baselines, I think it is necessary to clearly identify the differences between PARL and GAL in Sec. 4.2.
2. To my understanding, there are two differences between PARL and GAL as to the formulation of the loss function: 1) GAL takes the gradients of the loss while PARL takes the gradients of intermediate layers' outputs, and 2) GAL minimizes the (smoothed) maximum cosine similarity between pairs, while PARL minimizes the sum of the cosine similarity between pairs. An ablation study along these two axes would be very helpful for understanding which part is the key to the performance gain. Or at least, can the authors comment on the intuition/insight on these differences made by PARL?

### Weird trend in the results
1. The experimental results in general demonstrate the effectiveness of the proposed method. However, there is a weird trend in the results that raises my concern. In Table 1, when the dataset is CIFAR-100, the accuracy of PARL under the multi-step PGD and BIM (21.49% and 18.97%) is noticeably higher than the accuracy under the single-step FGSM (12.73%). To my knowledge the stronger PGD should always lead to lower accuracy than FGSM. I would like to see some clarifications from the authors on this.

### The choice of the layers
1. The paper presents ablation study as to the number of layers used to compute PARL loss function in Sec. 5.4. However, it remains unclear what is the motivation of choosing the first several layers. What if one uses the last several layers? What if one randomly select some candidates from the whole layers during each step?

**Summary Of The Paper:**

This work develops an ensemble-based adversarial defense which diversifies the sub-models to obtain robustness. The idea is to force the gradients of several layers pointing towards different directions across the sub-models such that the adversarial transferability can be reduced. On CIFAR-10/CIFAR-100, the paper demonstrates that PARL outperforms previous ensemble-based approaches.

**Summary Of The Review:**

Despite the better empirical results obtained by the proposed method, this work can be improved by identifying key difference to GAL and further investigate/justify the choice of the layers, which will help people understand why PARL is effective (currently it is not that clear). Also, I need some clarifications from the authors on the weird trend in the results. These concerns/issues currently prevent me from recommending acceptance.

---

### Official Review · Reviewer_BhHc · 2021-10-31

**Correctness:** 4
**Technical Novelty And Significance:** 2
**Empirical Novelty And Significance:** 2
**Recommendation:** 6
**Confidence:** 3

**Main Review:**

PRO:
- The paper is overall well written and easy to follow. The review of related work and the motivation of the proposal is well conveyed.
- The proposed regularizer is simple and easy to implement without too much burden. It is also nice authors attach the codes and even pre-trained models as supplementary materials.

CON:
However, I have to say the idea needs more refinement as the ablation study section does not provide too much information.
- Although the gradient of each layer of each classifier can be reused for regularization, will this slow down the computation? What will happen if we calculate the regularization more efficiently by sampling pairs of classifier?
- The gradient of each layer is calculated first in Section 4.1, and then used in the first equation of Section 4.2 (assign numbers for equations in revision). What may happen if the similarity is calculated feature-wised, instead of layer-wised? What if layers or features used for calculating similarity is sampled for computation efficiency?

**Summary Of The Paper:**

This paper propose a regularizer of layer-wise gradient to enhance the robustness against adversarial examples of an ensemble of deep neural network classifiers.

**Summary Of The Review:**

As explained in the main review, I admit the idea is good but it is at an early stage and needs more refinement.

---

### Official Review · Reviewer_Dge8 · 2021-11-02

**Correctness:** 3
**Technical Novelty And Significance:** 3
**Empirical Novelty And Significance:** 2
**Recommendation:** 3
**Confidence:** 5

**Main Review:**

Strengths
- Using ensembles with diverse members to increase robustness is an established and interesting topic.

- The method is a simple modification of standard training which doesn't incur in major additional cost and achieves some good result in the experimental evaluation.

Weaknesses
- In the proposed method the gradients, wrt the input, of the feature output of some layers are summed, and then the similarity of this sum with the corresponding one of the other classifiers is minimized: it is not clear to me which is the meaning of using the sum of the features of inner layers as a proxy of the classification loss. For example, if the output layer was used, the decision boundary would be represent by the difference between the logit of the correct class and the maximum of the others. Is this connected to the reason why only early layers are used? Moreover, in PARL the cosine similarity of pairs of models is minimized, but the minimal value -1 can be achieved only by a single pair. Does this create instability during training?

- Defending against black-box transfer attacks is a quite narrow scope. Moreover, the proposed method should, in my opinion, be compared to other techniques for robust (individual) models, first of all adversarial training, which should achieve, for example, around 48% robust accuracy at $\epsilon=0.031$ (CIFAR-10) against white-box attacks, see e.g. https://robustbench.github.io/, when using a small architecture. This appears to be better than what illustrated in Figure 3 for PARL.

- The strength of transfer attacks mostly depends on how similar the target and surrogate classifiers are: how was the surrogate ensemble used in the experiments trained? Moreover, since PARL uses majority voting to merge the output of the individual classifiers, how are the attacks carried out (see [A] for a discussion of this)?

- The evaluation appears to have some shortcomings: first, the worst-case, for each point, among different attacks should be reported. Second, the attacks used are very similar, that is gradient-based attacks on one loss (which, unless I missed it, is not specified): since the proposed method aims at diversifying the gradients across classifiers, it seems reasonable to test also an attack which doesn't rely on gradient [B, C] to generate the adversarial perturbations to be transferred. Moreover, I think it'd be important to know the success rate of the attacks on the surrogate models.

[A] https://arxiv.org/abs/2011.14031 \
[B] https://arxiv.org/abs/1912.00049 \
[C] https://openreview.net/forum?id=SygW0TEFwH

**Summary Of The Paper:**

The paper proposes a new method to train ensembles of classifiers which are robust to adversarial attacks, in particular black-box transfer-based ones. This is achieved by enforcing the output of early layers of different members of the ensemble to have, on average, gradients with low cosine similarity, which should in turn create different decision boundaries. For this, the authors design a specific loss function, PARL, to be minimized at training time. In the experiments, PARL models achieve higher robustness to transfer attacks that existing methods.

**Summary Of The Review:**

Overall, I think that the proposed method is not well-motivated. Also, the experiments do not include important baselines, and it is not clear how strong the evaluation used is (see above).

---

### Decision · Program_Chairs · 2022-01-20

**Decision:**

Reject

**Comment:**

The paper proposes a new method to train ensembles of classifiers that are robust to adversarial attacks, in particular black-box transfer-based ones. This is achieved by enforcing the output of early layers of different members of the ensemble to have, on average, gradients with low cosine similarity, which should in turn create different decision boundaries. For this, the authors design a specific loss function, PARL, to be minimized at training time. Two reviewers gave the score of 6 while two reviewers gave the score of 3. The main concerns are: 1) unclear meaning of taking the sum of the gradients of different neurons, and why the similarity of that across models is a proxy for similarity of the decision boundaries; 2) lack of experiments, that is, omitting a simpler baseline like individual robust classifiers. Positive score reviewers also did not champion the paper, thus, the paper should be well addressed these main concerns in the revision and cannot accept to ICLR for now.